# BitHydra: Towards Bit-flip Inference Cost Attack against Large Language Models

## Abstract

Large language models (LLMs) are widely deployed, but their growing compute demands expose them to inference cost attacks that maximize output length. We reveal that prior attacks are fundamentally *self-targeting* because they rely on crafted inputs, so the added cost accrues to the attacker's own queries and scales poorly in practice. In this work, we introduce the first **bit-flip inference cost attack** that directly modifies model weights to induce persistent overhead for all users of a compromised LLM. Such attacks are stealthy yet realistic in practice: for instance, in shared MLaaS environments, co-located tenants can exploit hardware-level faults (*e.g.*, Rowhammer) to flip memory bits storing model parameters. We instantiate this attack paradigm with `BitHydra`, which **(1)** minimizes a loss that suppresses the end-of-sequence token (*i.e.*, <EOS>) and **(2)** employs an efficient yet effective critical-bit search focused on the '<EOS>' embedding vector, sharply reducing the search space while preserving benign-looking outputs. We evaluate across 11 LLMs (1.5B–14B) under int8 and float16, demonstrating that our method efficiently achieves scalable cost inflation with only a few bit flips, while remaining effective even against potential defenses.

## 1 Introduction

Large Language Models (LLMs) (Carlini et al., 2021; Ouyang et al., 2022; Touvron et al., 2023) have demonstrated their remarkable capabilities across a wide range of real-world applications, including online chat (Shen et al., 2023), customer service (Gimpel et al., 2023), and financial services (Wu et al., 2023). As LLMs are increasingly deployed through cloud-based ML-as-a-Service (MLaaS) platforms, minimizing inference cost has become critical for both service providers and end-users—enhancing service availability and reducing token-based billing costs. However, previous studies have shown that deep neural networks are vulnerable to inference cost attacks (Shumailov et al., 2021; Shapira et al., 2022; 2023; Liu et al., 2023a; Schoof et al., 2024; Xiao et al., 2024; Ma et al., 2024; Müller & Quiring, 2024), where the attacker crafts malicious input to maximize the latency and cost of the victim model's inference execution. Such attacks can lead to substantial operational overhead for service providers and degrade the user experience. Recently, researchers designed inference cost attacks against auto-regressive LLMs (Feng et al., 2024; Geiping et al., 2024; Dong et al., 2024; Kumar et al., 2025) and multimodal LLMs (Gao et al., 2024). As the victim model's inference cost scales with the response length, the attacker's objective is to mislead the model to generate as many tokens as possible using short induced prompts.

Despite their diversity, existing inference cost attacks share a key feature: they rely on specially-crafted inputs to induce excessive computation. Consequently, this leads to two significant limitations in real-world scenarios. **(1)** These attacks are inherently *self-targeting*: the attacker, who submits the adversarial prompt, will be charged for the long generated responses, bearing the inference cost. **(2)** To achieve damages to other users and service providers at scale, the attacker needs to consistently send a large volume of malicious input, which can be costly and easy to spot.

We argue that the limitations of existing inference cost attacks primarily stem from their underlying threat model, in which the attacker is also the end-user and must therefore launch attacks solely through crafted inputs. Motivated by this observation, we propose a new class of inference cost attacks, termed bit-flip inference cost attacks (**BICAs**), which *target the model itself rather than its inputs*. The core idea is that flipping only a *few* critical weight bits can substantially increase

Figure 1: Comparison between traditional and bit-flip inference cost attacks. Traditional attacks, based on adversarial prompts, are self-targeting and affect only the attacker's queries. In contrast, our method modifies model weights (remotely), enabling persistent and widespread impact on all users interacting with the compromised model.

the computational cost for all subsequent queries, regardless of the user, without requiring any changes to the input, as illustrated in Figure 1. These attacks are plausible in various real-world scenarios where attackers can covertly tamper with model parameters. For example, a malicious tenant sharing the same cloud-based Machine Learning as a Service (MLaaS) platform may co-locate with the victim model on the same physical machine and exploit hardware-level vulnerabilities, *e.g.*, Deephammer (Yao et al., 2020), to flip critical bits in the model's weights *without physically touching the hardware device*. Such manipulations operate at the hardware level, and thus remain undetectable by conventional software-based monitoring or defenses.

However, implementing such BICAs introduces several technical challenges, including: **(1) Effectiveness**: how to design an effective loss function that encourages LLMs to generate substantially longer outputs; **(2) Scalability** how to efficiently identify the critical weight bits that significantly impact inference cost, given the vast number of parameters in LLMs; **(3) Fidelity**: how to ensure that, even after flipping these critical bits, the victim model continues to produce outputs that appear benign and exhibit no obvious anomalies. To tackle these challenges, we propose a simple yet effective attack method, dubbed `BitHydra`. Specifically, to achieve high attack effectiveness, we introduce a loss function $\mathcal{L}_{\texttt{<EOS>}}$, which penalizes the probability of output termination by suppressing the normalized likelihood of the end-of-sequence ($\texttt{<EOS>}$) token. Intuitively, minimizing $\mathcal{L}_{\texttt{<EOS>}}$ encourages the victim LLM to avoid generating the $\texttt{<EOS>}$ token, thereby producing abnormally long outputs without substantially impairing its general functionality. To overcome the scalability and fidelity challenges, `BitHydra` further incorporates a lightweight and efficient *critical bit search* algorithm. Instead of exhaustively searching across all model parameters, the algorithm strategically restricts the search to the output embedding layer and, more specifically, to the vector corresponding to the $\texttt{<EOS>}$ token. This method significantly reduces the search space, enabling rapid identification of high-impact bits. Simultaneously, by altering only a small and isolated portion of the model without affecting broader language representations, `BitHydra` preserves the victim model's ability to generate benign-looking content. This facilitates stealthy and persistent attacks that impose significant computational overhead while maintaining the normal utility and functionality of model responses.

In summary, our main contributions are four-fold. **(1)** We revisit existing inference cost attacks and reveal their inherent limitations and underlying reasons. **(2)** Based on our findings, we propose a new inference cost attack paradigm, *i.e.*, bit-flip inference cost attack (BICA), that targets model parameters rather than inputs, allowing large-scale persistent attacks that affect all users. **(3)** We design `BitHydra`, a simple yet effective instantiation of BICA that suppresses the occurrence of the end-of-sequence token with a few carefully chosen bit flips. **(4)** We demonstrate the effectiveness of `BitHydra` through extensive experiments, showing that it causes 100% of evaluation prompts to reach the maximum generation length on representative LLMs like Llama3-8B, while requiring as few as three bit flips in some cases. We also demonstrate `BitHydra`'s transferability to unseen prompts, suggesting a generalizable and systemic shift in generation dynamics.

## 2 BACKGROUND AND RELATED WORK

We present the background of inference cost attacks and BFAs in this section. Additional information about LLM and its data representation can be found in Appendix A.

### 2.1 INFERENCE COST ATTACKS

Inference cost attacks aim to exploit the compute-intensive nature of deep learning models to intentionally increase the models' latency or resource consumption during inference, ultimately leading to high compute cost and degraded user experience. Shumailov et al. (2021) introduced the

concept of *sponge examples* and designed the first inference cost attack. Later works extended this attack across various tasks and domains, such as image understanding (Chen et al., 2022b), object detection (Xiao et al., 2024; Ma et al., 2024), and language translation (Chen et al., 2022a).

Recent studies showed that this inference-cost threat is amplified in LLMs. LLMEffiChecker (Feng et al., 2024) employed gradient-guided search to find minimal, imperceptible input perturbations that raise inference cost; Geiping et al. (2024) coerced LLMs into generating specific starting responses, indirectly imposing higher computational cost; Dong et al. (2024) designed adversarial prompts that prolong decoding in modern autoregressive LLMs; Gao et al. (2024) crafted verbose images that elevate latency and energy use in multimodal LLMs; and Kumar et al. (2025) intentionally induced model 'overthinking', slowing its reasoning process.

However, to our best knowledge, all existing inference cost attacks induce damage solely by manipulating the model's inputs, which leads to two practical limitations. First, modern LLM services use token-based billing; for example, OpenAI's o3 API charges $10 per 1 million input tokens and $40 per 1 million output tokens (OpenAI, 2025). Thus, while abnormally long outputs increase the provider's computational load, the attacker ultimately pays the bill, and the provider suffers only mild externalities. Second, each adversarial input affects only its own inference, offering no persistent, cross-user impact. These limitations substantially reduce the practical severity of such attacks.

## 2.2 BIT-FLIP ATTACKS VIA ROWHAMMER

Bit-flip attacks (BFAs) are hardware-level attacks that tamper with critical bits in DRAM. A prominent vector is Rowhammer (Kim et al., 2014a), which rapidly activates aggressor rows to disturb adjacent cells and flip bits, even in the presence of common error-correction schemes (Gruss et al., 2018; Cojocar et al., 2019). Crucially, such faults can be triggered *without* physical access to the device, by running malicious code that repeatedly hammers memory on commodity CPUs (Jattke et al., 2022; Kogler et al., 2022) and GPUs with GDDR5 (Lin et al., 2025) or HBM (Olgun et al., 2024)

In the context of machine learning, attackers apply BFAs to flip selected bits in the parameters of a deployed model. Existing attacks are commonly categorized by the objectives: untargeted attacks (Rakin et al., 2019; Chen et al., 2023; Li et al., 2024) degrade overall model performance, whereas targeted attacks (Dong et al., 2023; Coalson et al., 2024) steer a model's behavior in specific ways, such as forcing misclassification or overriding content filters. To achieve precise and effective bit flips, attackers commonly pair Rowhammer with system-level memory placement tricks that rely on legitimate operating system features, *e.g.*, leveraging the page cache (Li et al., 2024), memory deduplication (Razavi et al., 2016), or per-CPU page-frame caches (Rakin et al., 2022). For instance, by first ensuring the model weights are resident in DRAM via the page cache and then inducing flips in those pages, attackers corrupt the in-memory copy so that subsequent loads by the victim process transparently retrieve the tampered weights from memory rather than the pristine file on disk.

Despite substantial progress, BFAs have been studied primarily on DNNs, and their feasibility for LLMs remains largely underexplored. More importantly, prior work targets accuracy degradation or specific misbehavior, which differs fundamentally from our objective: inflating the computational cost of ordinary queries while preserving task correctness. Existing BFA techniques are ill-suited for this purpose as they do not identify weight bits that modulate execution paths or computational complexity. Designing an effective bit-flip inference-cost attack for LLMs remains an open problem.

## 3 PROBLEM FORMULATION AND ANALYSIS

To address the limitations of traditional inference-cost attacks, we shift from prompt perturbations to weight manipulation via bit-flip techniques. In our paradigm, an adversary flips a small set of cost-critical bits in a target LLM, biasing it to produce longer responses to *any* prompt and thereby scaling provider-side computation. We term this the bit-flip inference-cost attack (BICA). This section formalizes the threat model and analyzes the key challenges in realizing BICA.

### 3.1 THREAT MODEL

**Attacker's Goal**. The attacker aims to inflate the inference cost of a deployed LLM persistently and at scale, without compromising task accuracy. Specifically, the objective is to induce the model to generate abnormally long responses for *any* user prompt, thereby amplifying computational

overhead across users and sessions. This shifts the attack surface from input manipulation to weight manipulation, enabling broad, cross-user impact that extends beyond the attacker's own queries. Such an attack poses serious risks to both users and LLM-integrated service providers. For users, it leads to elevated query costs and significantly increased latency, particularly detrimental for time-sensitive applications, ultimately degrading the overall user experience. For providers, the attack escalates operational costs and may cause query congestion due to slower response time. Over time, reduced service availability and increased expenses may finally lead to user attrition and reputational harm.

**Attacker's Capacity**. This paper adopts the same threat model (Rakin et al., 2019; Yao et al., 2020; Liu et al., 2023b; Li et al., 2024) in the conventional BFAs. Specifically, the target LLM runs in a resource-sharing environment (*e.g.*, an MLaaS platform), and the adversary is an unprivileged co-located tenant on the same physical machine as the victim. The adversary can induce bit flips in DRAM via Rowhammer (Kim et al., 2014b) *without physical access or elevated privileges*. Besides, the attacker does not possess training data but has white-box knowledge of the model's architecture and weights. Arguably, this setting is both practical and commonly seen in real-world scenarios (Meta, 2024; AWS, 2024; Microsoft, 2025), as many companies and application developers deploy open-source LLMs on public cloud platforms (*e.g.*, AWS and Azure) for high scalability, flexible deployment, and convenient access to powerful GPU resources.

### 3.2 MAIN CHALLENGES OF INSTANTIATING BICA

Building a bit-flip inference cost attack imposes more strict constraints than traditional accuracy-degrading BFAs. A successful method must **(1)** inflate the output length under normal usage while **(2)** preserving functional plausibility **(3)** under a very small flip budget to remain practical and stealthy. In our early exploration, we attempted a brute-force strategy that scans the entire weight space for all potential bits and measures their effects. This naive approach revealed three fundamental obstacles: catastrophic numerical failures, visible degradation of linguistic quality, and prohibitive search cost, which collectively motivate a structure-aware design introduced by our method.

**Challenge 1 (Catastrophic Numerical Failures)**. Flipping arbitrary bits frequently drives the LLM into catastrophic model states, with decoding collapsing to 'NaN' after only a few flips in many cases. This failure mode is uncommon in traditional BFAs in attacking feedforward CNN/MLP settings but is amplified in LLMs due to their autoregressive nature and tightly coupled operations (*e.g.*, LayerNorm, Softmax, attention scaling) over long sequences. A perturbed early-layer weight can be magnified through normalization and exponentiation, triggering overflow/underflow or near-zero variance divisions; the instability then recurs across decoding steps, culminating in the NaN outputs.

**Challenge 2 (Visible Degradation of Linguistic Quality)**. Even when the model does not crash, bit-flipping often yields incoherent text, such as garbled symbols, broken tokens, and non-linguistic artifacts. This indicates that bit flips scattered throughout the model can disrupt semantic and syntactic alignment, degrading internal representations beyond recovery. Unlike vision models, where spatial redundancies/correlations can buffer mild corruption, LLMs lack comparable structural slack, so small weight modifications can visibly erode linguistic fidelity. A successful BICA must therefore identify critical bits that lengthen the output while preserving generation plausibility and task utility.

**Challenge 3 (Prohibitive Search Cost)**. Exhaustively scanning and evaluating bits in large-scale LLMs (*e.g.*, with billions of parameters) is computationally prohibitive. Loading full weight matrices for gradient- or search-based scoring, running per-flip impact tests, and measuring downstream cost inflation impose heavy memory and latency overheads. A viable BICA requires an efficient search strategy that narrows the candidate space and prioritizes *cost-critical* locations, achieving persistent cost inflation with a small flip budget.

## 4 METHODOLOGY

### 4.1 OVERALL WORKFLOW

Guided by the analysis in the previous section, we design `BitHydra` to achieve a practical and scalable inference cost attacks via bit flips. Its key idea is to design a loss function that penalizes the normalized probability of generating the `<EOS>` token and effectively reduces its value by flipping the critical bits of the victim LLM's parameters. To ensure stability, fidelity, and efficiency, we constrain flips to the single row of the output embedding matrix corresponding to `<EOS>`. This targeted scope avoids perturbing intermediate layers (mitigating numerical instability), preserves

Figure 2: Overview of `BitHydra`. BitHydra consists of three stages: (1) **Significant Weight Identification**: Attackers identify significant weights within the <EOS> token's embedding row guided by the loss that penalizes the probability of generating the <EOS> token; (2) **Target Bit Selection**: Attackers select the bit flips needed to approximate the target weight changes; and (3) **Bit Flipping**: attackers use Rowhammer to remotely induce the selected bit errors in DRAM.

normal token logits (maintaining linguistic plausibility), and drastically shrinks the search space to a handful of high-impact weights, directly addressing the challenges identified earlier.

In general, as shown in Figure 2, our `BitHydra` operates in three stages: **(1)** Significant Weight Identification, **(2)** Target Bit Selection, and **(3)** Bit Flipping. Stages 1-2 are performed offline, while Stage 3 is carried out online. Specifically, in the first stage, we analyze the output embedding row corresponding to the <EOS> token and identify weights that most influence the model's tendency to terminate generation. This is achieved by optimizing the proposed loss function $\mathcal{L}_{\text{<EOS>}}$ on a set of prompts to find weights whose perturbation significantly lowers the likelihood of generating <EOS>; In the second stage, for each selected weight, we determine the most effective bit index to flip so that the resulting value approximates the optimized target, minimizing deviation while maximizing impact; In the last stage, the attacker executes the bit-level perturbations using Rowhammer-based techniques. This involves memory profiling (Pessl et al., 2016) to identify vulnerable DRAM cells, memory massaging (Kwong et al., 2020) to align these cells with target bits, and controlled hammering to induce the desired bit flips in memory.

In particular, because the mechanics of the third stage can be implemented with well-established Rowhammer techniques (Yao et al., 2020), the remainder of this section concentrates on the first two stages: the design of $\mathcal{L}_{\text{<EOS>}}$ and the efficient search for cost-critical weights and bits.

### 4.2 STAGE 1: SIGNIFICANT WEIGHT IDENTIFICATION

Given the target LLM, the attacker first identifies a subset of weights in the output embedding layer whose perturbations most effectively suppress the termination signal (*i.e.*, <EOS> token) and thereby extend generation length. The selection is based on gradient analysis: in each search round, we evaluate the gradient magnitudes of our pre-defined loss function $\mathcal{L}_{\text{<EOS>}}$ and flip a single bit in the weight corresponding to the maximum gradient value. More details are as follows.

**Loss Design for Early Termination Suppression.** To encourage prolonged generation, we define a loss function $\mathcal{L}_{\text{<EOS>}}$ that penalizes the probability of output termination by suppressing the normalized likelihood of the end-of-sequence (<EOS>) token over the entire generation sequence:

$$\mathcal{L}_{\text{<EOS>}}(\boldsymbol{x}) = \sum_{i=1}^{N} \text{Softmax}(f_i^{\text{<EOS>}}(\boldsymbol{x})), \tag{1}$$

where $f_i^{\text{<EOS>}}(\cdot)$ denotes the logit assigned to the <EOS> token at step $i$, and $N$ is the total number of decoding steps. In particular, we hereby use the normalized probability instead of raw logits to better capture the relative likelihood of <EOS> in context. More discussions are in Section 5.3.

**Gradient Ranking to Identify Significant Weights.** Given $\mathcal{L}_{\text{<EOS>}}$, we seek to identify the weights that most significantly impact termination suppression. Specifically, in each search round, we compute the gradient of $\mathcal{L}_{\text{<EOS>}}$ with respect to the output embedding layer $\boldsymbol{W}_o$, which maps the decoder hidden state $\boldsymbol{h} \in \mathbb{R}^d$ to the vocabulary logits $\boldsymbol{l} \in \mathbb{R}^V$.

We hereby restrict updates solely to the row $\boldsymbol{W}_o[\text{<EOS>}] \in \mathbb{R}^d$, corresponding to the <EOS> token, since our objective is to reduce the probability of this specific token without affecting the rest of the vocabulary. Arguably, updating only $\boldsymbol{W}_o[\text{<EOS>}]$ ensures minimal interference with generation quality and semantic coherence.

The accumulated gradient matrix for one epoch is:

$$\hat{G} = \frac{\partial \mathcal{L}_{\texttt{<EOS>}}}{\partial W_o} = \begin{array}{c} \text{OUT}_1 \\ \vdots \\ \text{OUT}_{\texttt{<EOS>}} \\ \vdots \\ \text{OUT}_V \end{array} \begin{bmatrix} \begin{array}{ccc} \text{IN}_1 & \cdots & \text{IN}_d \\ g_{1,1} & \cdots & g_{1,d} \\ \vdots & \ddots & \vdots \\ g_{\texttt{<EOS>},1} & \cdots & g_{\texttt{<EOS>},d} \\ \vdots & \ddots & \vdots \\ g_{V,1} & \cdots & g_{V,d} \end{array} \end{bmatrix}, \tag{2}$$

and the update step is defined as:

$$W_o[\texttt{<EOS>}] = W_o[\texttt{<EOS>}] - \texttt{scale}\left(\hat{G}[\texttt{<EOS>}]\right), \tag{3}$$

where only the gradient row $\hat{G}[\texttt{<EOS>}]$ is used for the update; all other rows of $W_o$ are preserved.

**Dynamic Gradient Normalization.** Unlike conventional training regimes, our loss function $\mathcal{L}_{\texttt{<EOS>}}$ is large at the beginning, but decreases rapidly after a few epochs, often resulting in vanishing gradients. To mitigate this issue, we introduce a dynamic function $\texttt{scale}$ that normalizes the gradient magnitude: if the $\ell_2$-norm of $\hat{G}[\texttt{<EOS>}]$ falls outside of a predefined range $[\texttt{grad}_{low}, \texttt{grad}_{up}]$, it is rescaled into this interval. It maintains efficacy while preventing instability due to small gradients. After gradient computation, we rank the absolute gradient magnitudes to identify critical weights:

$$\text{Top}_n\left(\left|[g_{\texttt{<EOS>},1}, g_{\texttt{<EOS>},2}, \dots, g_{\texttt{<EOS>},d}]\right|\right), \tag{4}$$

where $n$ is the number of allowed bit flips. This selects the top-$n$ dimensions with the largest absolute gradients, whose corresponding updated values are passed to the next stage.

**Functional Stealthiness via Localized Modification.** This targeted modification of $W_o[\texttt{<EOS>}]$ ensures minimal disruption to the model's generation dynamics. To justify this, consider the perturbed logit vector $l'$, where

$$l'(i) = \begin{cases} (W_o[\texttt{<EOS>}] + \Delta W) \cdot h, & \text{if } i = \texttt{<EOS>} \\ W_o[i] \cdot h, & \text{otherwise} \end{cases}, \tag{5}$$

and $\Delta W$ is the perturbation vector. Since all logits for $i \neq \texttt{<EOS>}$ remain unchanged, the Softmax-normalized relative ranking among normal tokens is preserved:

$$\frac{P(i)}{P(j)} = \frac{e^{l'(i)}}{e^{l'(j)}} = \frac{e^{l(i)}}{e^{l(j)}}, \quad \forall i, j \neq \texttt{<EOS>}. \tag{6}$$

Only the ranking of the $\texttt{<EOS>}$ token is altered due to the modified logit. As such, the model continues to generate coherent and fluent content, while the probability of termination is suppressed.

**Attack Interpretation.** To further explain the effectiveness, we analyze how the perturbation to the $\texttt{<EOS>}$ token weight vector $W_o[\texttt{<EOS>}]$ affects its interaction with the model's hidden representations. Recall that the logit for the $\texttt{<EOS>}$ token at each decoding step is computed as the dot product between $W_o[\texttt{<EOS>}]$ and the hidden state $h \in \mathbb{R}^d$, i.e., $l_{\texttt{<EOS>}} = W_o[\texttt{<EOS>}] \cdot h$. A reduction in this logit can arise from either a smaller norm of $W_o[\texttt{<EOS>}]$ or a decreased alignment between $W_o[\texttt{<EOS>}]$ and $h$. We measure the *cosine similarity* between $W_o[\texttt{<EOS>}]$ and $h$ at each decoding step, before

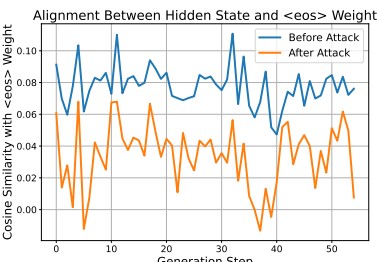

Figure 3: Cosine similarity at each steps.

and after the attack. As shown in Figure 3, the cosine similarity significantly decreases across the entire generation process after we flip the identified bits. This is a clear indication that the modified $W_o[\texttt{<EOS>}]$ is no longer aligned with the hidden states that typically trigger the sequence termination. This explains the drop in the $\texttt{<EOS>}$ probability and thus the extension of output length, without affecting other tokens whose logits remain unchanged.

### 4.3 STAGE 2: TARGET BIT SELECTION

For each identified weight $W_o^i$, the attacker selects the optimal bit position(s) within the weight value to flip, such that the flipped weight is as close as possible to the target value $W_o'^i$ produced in the

first stage. Taking a single bit-flip as an example, the goal is to approximate the target weight using a single-bit flip in the original weight $\boldsymbol{W}_o^i$, as follows:

$$b^* = \arg \min_{b \in \{0,\ldots,B-1\}} \left| \mathrm{Fp}(\mathrm{FlipBit}(\boldsymbol{W}_o^i, b)) - \boldsymbol{W}_o'^i \right|, \qquad (7)$$

where $B$ is the number of bits in the data type (*e.g.*, $B = 8$ for int8), $\mathrm{FlipBit}(\boldsymbol{W}_o^i, b)$ returns the binary representation of $\boldsymbol{W}_o^i$ with the $b$-th bit flipped, and $\mathrm{Fp}(\cdot)$ converts the resulting binary back into its floating-point equivalent.

For the int8 data format, we traverse all 8 bits in each weight and flip them one by one to evaluate the effect of each flip. The bit that results in the closest absolute value to the target weight is selected. A quantization scale factor $F$ is used to convert between the quantized integer value $int_{\mathrm{weight}} \in [-128, 127]$ and its corresponding floating-point value $fp_{\mathrm{weight}} \in [-F, F]$, following the relation $fp_{\mathrm{weight}} = int_{\mathrm{weight}} \times F/127$. A similar procedure is applied to the float16 format, taking into account its internal bit layout, including sign, exponent, and mantissa components.

Note that the process described above solely identifies a *single* optimal bit to flip for a given weight. To perform *multi-bit* flipping within the same weight, the procedure can be repeated iteratively: after flipping one bit, the weight is updated, and a new target can be defined to guide the next bit selection. The full algorithm is provided in Appendix B.

**Progressive v.s. One-shot Search.** BitHydra supports two modes (*i.e.* Progressive and One-shot) when flipping multiple bits. In the *one-shot* mode, all critical weights are selected and their bit flips are determined in a single pass. In contrast, the *progressive* mode iteratively identifies and flips the most critical bit in the most important weight during each round. After applying each flip, the search continues based on the updated model state. One-shot search is substantially more time-efficient because it completes in a single loop, whereas progressive search better captures cumulative interactions among flips and can flip multiple distinct bits within the same weight across rounds (one-shot mode can flip at most one bit per weight).

Our experiments indicate that, under int8 quantization, progressive and one-shot searches obtain similar attack effectiveness, but one-shot is markedly faster and thus preferred. We attribute this to the limited representable range in int8: the maximum effective change from a bit flip is bounded by the scale of the largest weight, constraining the realized impact of theoretically optimal refinements. Consequently, progressive refinement offers limited practical advantage, and a simpler one-shot approach suffices. On the other hand, in the float16 setting, progressive search generally achieves better results. Since float16 provides a much wider and finer-grained representable range, progressive updates can more effectively leverage accumulated small changes over multiple rounds to induce stronger attack effects. In summary, one-shot search is preferred for quantized models due to its speed and comparable effectiveness, while progressive search is more effective for high-precision formats like float16 where bit-level manipulations have finer resolution and stronger cumulative impact.

## 5 EVALUATION

### 5.1 EXPERIMENTAL SETTINGS

**Models and Datasets.** We evaluate on 11 LLMs across six families: DeepSeek-R1-Distill-Qwen (1.5B) (DeepSeek-AI, 2025), Qwen1.5 (1.8B and 4B) (Bai et al., 2023), Samantha (7B) (sam, 2023), Vicuna (7B, v1.3 and v1.5) (Chiang et al., 2023), Llama-2-7b-chat-hf (lla, 2023), Mistral-Instruct (7B, v0.3) (Mis, 2024), Meta-Llama-3-Instruct (8B) (AI@Meta, 2024), DeepSeek-R1-Distill-Llama (8B) (DeepSeek-AI, 2025), and Qwen2.5-Instruct (14B) (Team, 2024). For each model, we test float16 (FP16) and int8 variants via (Dettmers et al., 2022). We adopt the Stanford Alpaca dataset (Standford, 2025) for both vulnerable-bit search and evaluation, adopting the first 100 instruction–response pairs as a common prompt set across all models.

**Baselines.** We compare against two categories. First, we replicate three prompt-based inference-cost attacks: **(1)** Engorgio (Dong et al., 2024), **(2)** LLMEffiChecker (Feng et al., 2024), and **(3)** Sponge Examples (Shumailov et al., 2021). Second, as no prior work applies BFAs directly to inference-cost attacks, we adapt Prisonbreak (Coalson et al., 2024) from jailbreak to our objective by replacing its loss with our end-of-sequence loss $\mathcal{L}_{\mathrm{<EOS>}}$. Following the original setting, this baseline permits flips across the *entire* model rather than restricting to the last layer as in BitHydra.

Table 1: Main attack results of our `BitHydra`. The maximum generation length is set to 2048.

| Model | Size (B) | AvgLen (Ori) | Int8 Attack Result | | | | Fp16 Attack Result | | | |
|---|---|---|---|---|---|---|---|---|---|---|
| | | | #Sample | #BitFlip | AvgLen | MaxRate | #Sample | #BitFlip | AvgLen | MaxRate |
| DeepSeek | 1.5 | 1117 | 4 | 8 | 1973 | 93% | 9 | 10 | 1968 | 96% |
| Qwen1.5 | 1.8 | 206 | 4 | 4 | 2047 | 98% | 4 | 7 | 2048 | 100% |
| Qwen1.5 | 4 | 254 | 4 | 12 | 2048 | 100% | 4 | 21 | 2026 | 96% |
| Samantha | 7 | 243 | 12 | 26 | 2048 | 100% | 4 | 21 | 2048 | 100% |
| Vicuna1.3 | 7 | 215 | 4 | 15 | 1990 | 94% | 9 | 5 | 1780 | 87% |
| Llama2 | 7 | 191 | 6 | 30 | 1880 | 90% | 6 | 17 | 2048 | 100% |
| Mistral | 7 | 250 | 4 | 14 | 2048 | 100% | 9 | 28 | 2048 | 100% |
| Vicuna1.5 | 7 | 226 | 4 | 25 | 1905 | 93% | 9 | 15 | 1628 | 80% |
| Llama3 | 8 | 260 | 4 | 3 | 2048 | 100% | 4 | 5 | 2048 | 100% |
| DeepSeek | 8 | 384 | 4 | 13 | 2021 | 96% | 4 | 3 | 2014 | 98% |
| Qwen2.5 | 14 | 265 | 4 | 7 | 2048 | 100% | 6 | 6 | 1990 | 96% |

Table 2: Comparison with baselines. The maximum output length is set to 2048 in these experiments.

| Attack Type↓ | Llama2-7B | | Samantha-7B | | Vicuna-7B | |
|---|---|---|---|---|---|---|
| | AvgLen | MaxRate | AvgLen | MaxRate | AvgLen | MaxRate |
| No Attack | 191 | 0% | 243 | 0% | 215 | 0% |
| LLMEffiChecker | 628 | 8% | 272 | 1% | 362 | 3% |
| Sponge examples | 457 | 15% | 1268 | 60% | 84 | 0% |
| Engorgio | 1856 | 89% | 1149 | 48% | 853 | 10% |
| Prisonbreaker | 712 | 28% | 1749 | 85% | 3 | 0% |
| BitHydra | **2048** | **100%** | **2048** | **100%** | **1780** | **87%** |

**Evaluation Metrics.** We assess effectiveness and efficiency using four metrics: **(1)** *AvgLen (Ori)*: average output length of the original LLM; **(2)** *AvgLen (Attack)*: average output length after bit flips; **(3)** *MaxRate*: fraction of outputs that hit the preset maximum generation length; and **(4)** *#BitFlip*: total number of flipped bits during attacks.

## 5.2 MAIN RESULTS

We present the main results; additional evaluation of the impact of output quality is in Appendix C.2.

**Performance across Different LLMs.** As shown in Table 1, our method demonstrates strong performance: with as few as 3–30 bit flips, `BitHydra` can significantly prolong the output generation. For most models, over 90% of user prompts reach the maximum generation length, and even 100% in several cases. The average response length approaches or hits the 2048-token cap. In the Int8 setting, which imposes tighter representation constraints than FP16, our attack still performs remarkably well, often requiring even fewer bit flips. This highlights the precision-agnostic nature of the vulnerability.

**Transferability to Unseen Prompts.** As shown in Table 1, in addition to high attack success rates, a crucial strength of our proposed attack lies in its strong *transferability*—the ability of bit flips computed using a few search prompts to generalize and induce unbounded output across a wide range of unseen inputs. For instance, in the case of the LLaMA3 8B model with `int8` quantization, using only 4 samples for gradient-based bit selection, the attack causes every prompt in a 100-prompt test set to generate until the maximum sequence length of 2048 tokens. To further assess this transferability, we compute the average cosine similarity between each of the 4 search prompts and the 100 test prompts in the Alpaca dataset using an embedding-based metric. The resulting average similarities for the 4 search prompts are 0.0818, 0.1125, 0.1151, and 0.0957, respectively. These relatively low similarity values indicate that the search and test prompts are semantically diverse. This reinforces the conclusion that the model's altered behavior is not the result of memorizing or overfitting to the search prompts, but rather reflects a generalizable and systemic shift in generation dynamics.

**Comparison with Baseline Attacks.** As shown in Table 2, across all tested models, our method consistently outperforms baselines in both average generation length and percentage of samples reaching the maximum token limit. Specifically, our approach achieves 100% MaxRate on LLaMA2-7B and Samantha-7B. In contrast, baseline attacks demonstrate uneven performance across models. Moreover, we observe that outputs generated under Prisonbreaker frequently contain meaningless symbols and non-linguistic artifacts. These observations support the point raised in Section 3.2: indiscriminately flipping bits across the entire model can lead to catastrophic and unpredictable outcomes—both in terms of functional degradation and unintended behaviors.

**Additional Attack Surface**. Example outputs from `BitHydra`-affected models appear in Appendix D. In several cases, prolonged generation inadvertently revealed internal system prompts or hidden metadata that should remain confidential. This unintended leakage underscores a novel and concerning attack surface (Li et al., 2025) , which we leave for further investigation.

Table 3: Ablation study of loss aggregation strategy.

| Agg. Type↓ | Qwen1.5-1.8B | | Llama-3-8B | | DeepSeek-R1-8B | |
|---|---|---|---|---|---|---|
| | AvgLen | MaxRate | AvgLen | MaxRate | AvgLen | MaxRate |
| Full | **2048** | **100%** | **2048** | **100%** | **2014** | **98%** |
| Latter Half | 2012 | 98% | 1987 | 96% | 1646 | 69% |
| Last | 1902 | 87% | 1987 | 96% | 440 | 2% |

Table 4: `BitHydra`'s resistance to possible defenses.

| Defense↓ | Qwen1.5-1.8B | | Llama-3-8B | | DeepSeek-R1-8B | |
|---|---|---|---|---|---|---|
| | AvgLen | MaxRate | AvgLen | MaxRate | AvgLen | MaxRate |
| None | 2048 | 100% | 2048 | 100% | 2014 | 98% |
| Fine-tuning | 2046 | 98% | 2022 | 98% | 1984 | 98% |
| Weight Recon. | 2023 | 96% | 2022 | 98% | 1299 | 50% |

## 5.3 ABLATION STUDY

We hereby evaluate `BitHydra` under different loss functions, where the optimal settings are **bolded** in Table 1 for comparison. Additional ablation studies on gradient scaling, search-sample count, and decoding temperature are provided in Appendix C.3.

**Impact of Loss Aggregation Strategy.** `BitHydra` employs a customized loss (*i.e.*, $\mathcal{L}_{\texttt{<EOS>}}$) that accumulates the probability of generating `<EOS>` across decoding. By default, we aggregate over all steps to capture the model's overall termination tendency. To evaluate this choice, we compare three strategies: **(1)** sum over the full sequence, **(2)** sum over only the latter half, and **(3)** use only the final step. Table 3 shows that full-sequence aggregation is crucial, consistently achieving the highest MaxRate (94–100%) and the lowest AvgLen, indicating that early steps provide valuable gradients for identifying effective bit flips.

## 5.4 RESISTANCE TO POTENTIAL DEFENSES

**Settings**. Existing model-level defenses against malicious bit flips generally fall into two main categories: *detection-based* (Javaheripi & Koushanfar, 2021; Li et al., 2021; Chen et al.; Javaheripi et al., 2022) and *prevention-based* (Li et al., 2020; He et al., 2020; Chen et al., 2021) approaches. Detection methods monitor inference to flag and recover from flip-induced errors but often incur substantial overhead—especially on LLMs (Coalson et al., 2024). We therefore evaluate `BitHydra`'s robustness against two representative *prevention* strategies: **(1)** *fine-tuning* to perturb the locations of previously identified critical bits (Wang et al., 2023) via LoRA on the full Alpaca training set for 3 epochs, and **(2)** *weight reconstruction* to reduce bit-level sensitivity (Li et al., 2020) via per-layer clipping to original min/max values at inference.

**Results**. As shown in Table 4, with fine-tuning, the attack remains highly effective across all three models: AvgLen slightly increases or remains stable, and MaxRate declines only marginally (2–4%). Weight reconstruction yields mixed outcomes: for Llama3-8B and Qwen1.5-1.8B it is largely ineffective (MaxRate 96–98%, with negligible AvgLen change), but it shows partial efficacy on DeepSeek-R1-8B. This suggests that DeepSeek's perturbations are more tightly constrained, potentially due to weight distribution or output-layer sensitivity, so when adversarial flips push weights outside the model's original clipping bounds, the defense can neutralize them more effectively. These results verify the resistance of our `BitHydra` to potential defenses.

## 6 CONCLUSION

This work presented `BitHydra`, a novel bit-flip inference cost attack against LLMs. Unlike prompt-based methods that increased latency via crafted inputs, we corrupted model weights to induce persistent, cross-user cost inflation. We instantiated this strategy with a loss that suppressed the likelihood of the end-of-sequence token (*i.e.*, `<EOS>`) and an efficient critical-bit search confined to the `<EOS>`-embedding row, enabling a few targeted flips to prolong generation while preserving output plausibility. Extensive experiments across diverse LLMs and precisions showed that `BitHydra` achieved scalable cost inflation with a few flips and remained effective under potential defenses. These findings exposed a significant yet underexplored threat surface, underscoring the need for routine weight-integrity monitoring and deployment- and inference-time safeguards in LLM services.

ETHICS STATEMENT

This work highlighted a critical and previously underexplored vector of inference cost attacks against large-scale language models through parameter-level manipulation. All experiments were conducted in controlled research environments, and no commercial systems were targeted or harmed. By demonstrating how small bit-level changes could significantly affect model behavior, we aimed to inform practitioners and developers about the potential risks of deploying LLMs in untrusted environments, such as shared MLaaS environments. Our `BitHydra` facilitated the study of this threat surface and provided insights that could support the development of stronger hardware and software safeguards, such as integrity verification mechanisms and parameter corruption detection tools, ultimately leading to more secure and reliable AI systems. As with many security-oriented contributions, we acknowledged that the methodology could in principle be misused. For example, an attacker with memory access could attempt to deploy such bit-flip attacks to degrade system availability or inflate operational costs. Nonetheless, we believed that the benefits of exposing this class of vulnerabilities for the purpose of building effective defenses outweighed the risks of potential misuse. Importantly, although our method demonstrated resilience against representative defenses, developers could still mitigate such threats fundamentally by deploying models only in trusted environments, enforcing regular integrity checks, and adopting tamper-resistant hardware or secure memory architectures. We advocated for responsible model deployment practices and would further explore defense strategies against such attacks in our future work.

REPRODUCIBILITY STATEMENT

Details of our implementation and experimental setup are provided in C.1. We include the inference code for `BitHydra` and instructions for running it in the supplementary material. The complete codebase, including the bit search procedure, will be released upon acceptance of the paper.

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

# A  BACKGROUND

## A.1  LARGE LANGUAGE MODELS (LLMs)

Large Language Models (LLMs) are typically built upon the decoder-only Transformer architecture Vaswani et al. (2017). Its autoregressive nature supports sequential token prediction conditioned on past context. Formally, given an input token sequence $\mathbf{x} = (x_1, x_2, \ldots, x_T)$, the model aims to estimate the joint probability by chaining conditional probabilities:

$$P(\mathbf{x}) = P(x_1) \cdot \ldots \cdot P(x_T \mid x_{1:T-1}) = \prod_{t=1}^{T} P(x_t \mid x_{<t}),$$

where $x_{<t}$ represents the prefix subsequence $(x_1, \ldots, x_{t-1})$.

An LLM can be abstracted as a function $f_\theta : \mathbb{Z}^t \to \mathbb{R}^V$, which maps a sequence of token IDs to a logit vector $\mathbf{z}_t = f_\theta(x_1, \ldots, x_t)$, where $V$ is the size of the vocabulary. At each decoding step, the LLM outputs a distribution over the next token. The generation process is typically initialized with a special start token (<sos>), and proceeds iteratively—appending new tokens to the input—until either the end-of-sequence token (<EOS>) is produced or a predefined maximum length is reached.

## A.2  DATA REPRESENTATION IN LLMs

As language models grow in size, the demand for memory and compute efficiency becomes critical. To this end, modern LLMs often adopt lower-precision numerical formats instead of the conventional 32-bit single-precision floating-point (fp32). Common formats include 16-bit half-precision floating-point (fp16), 8-bit integers (int8), 4-bit integers (int4), and 4-bit normalized floating-point (nf4). They help reduce the memory footprint and improve inference speed. In this paper, we mainly focus on the int8 and fp16 formats, which are widely used in real-world deployment.

**Int8 Data Format.** Each layer's weight tensor is scaled and rounded to fit into an 8-bit integer representation. Specifically, for the $l$-th layer, the quantization process can be described as:

$$\Delta w_l = \frac{\max(|\mathbf{W}_l|)}{2^7 - 1}, \quad \mathbf{W}_l \in \mathbb{R}^d \tag{8}$$

$$\mathbf{W}_l^q = \text{round}\left(\frac{\mathbf{W}_l}{\Delta w_l}\right) \cdot \Delta w_l \tag{9}$$

where $d$ is the number of weights in layer $l$, $\Delta w_l$ is the quantization step size, $\mathbf{W}_l$ is the original weight tensor, and $\mathbf{W}_l^q$ is the quantized version.

In computer systems, signed integers are typically represented using two's complement encoding. For a quantized weight $w/\Delta w$ represented by an 8-bit binary vector $\mathbf{b} = [b_7, b_6, \ldots, b_0] \in \{0,1\}^8$, its value is reconstructed as:

$$\frac{w}{\Delta w} = -2^7 \cdot b_7 + \sum_{i=0}^{6} 2^i \cdot b_i \tag{10}$$

Several efficient quantization libraries such as BitsAndBytes Dettmers et al. (2022) support multiple schemes for implementing int8 quantized weights in LLMs.

**FP16 Data Format.** Weights stored in this format follow the IEEE 754 half-precision floating-point standard. Each value is represented using 16 bits: 1 sign bit ($s$), 5 exponent bits ($e$), and 10 mantissa (fraction) bits ($m$). The actual weight value $w$ represented by an FP16 number is computed as:

$$w = (-1)^s \cdot 2^{(e-15)} \cdot \left(1 + \frac{m}{2^{10}}\right) \tag{11}$$

FP16 significantly reduces the memory footprint while retaining a sufficient dynamic range and precision for most deep learning applications.

# B  PESUDO CODE OF BIT FLIPPING.

---

**Algorithm 1** Bit flipping in target weights

---

1: **Input:** *WeightDict:* $\{(i, W_o^i, W_o^{'i})\}$, *DType*, $F \triangleright$ Top-$n$ weights: index $i$, original weight $W_o^i$, target weight $W_o^{'i}$; data type; quantization scale factor
2: **Output:** *FlipDict* $\hspace{5cm} \triangleright$ Bit flip positions
3: FlipDict $\leftarrow \emptyset$
4: **for** $(i, W_o^i, W_o^{'i}) \in$ WeightDict **do**
5: $\quad$ BestBit $\leftarrow$ None
6: $\quad$ BestWeight $\leftarrow$ None
7: $\quad$ FpWeight $\leftarrow$ ConvertToFp($W_o^i$, DType, $F$)
8: $\quad$ BinWeight $\leftarrow$ ConvertToBin($W_o^i$, DType)
9: $\quad$ **for** bit $= 0$ **to** DType.bitlength$-1$ **do**
10: $\quad\quad$ FlippedBinWeight $\leftarrow$ FlipBit(BinWeight, bit)
11: $\quad\quad$ FlippedFpWeight $\leftarrow$ ConvertToFp(FlippedBinWeight, DType, $F$)
12: $\quad\quad$ **if** $|$FlippedFpWeight $- W_o^{'i}| < |$BestWeight $- W_o^{'i}|$ **then**
13: $\quad\quad\quad$ BestBit $\leftarrow$ bit
14: $\quad\quad\quad$ BestWeight $\leftarrow$ FlippedFpWeight
15: $\quad$ FlipDict.append(($i$, BestBit))

---

# C  ADDITIONAL EVALUATION

## C.1  TESTBED

We conduct our experiments on NVIDIA GeForce RTX 3090 GPUs, GeForce RTX 4090D GPUs, and RTX A6000 GPUs. The software environment includes CUDA version 12.4, Transformers version 4.48, and PyTorch version 2.0.1. On a 4090D GPU, the one-shot search process takes approximately 4 minutes for a 7B `float16` model. The progressive search requires about 5 minutes per bit flip for the same model and hardware configuration.

## C.2  IMPACT ON OUTPUT QUALITY

To evaluate whether flipping EOS-related weights leads to degradation in output quality, we assess the generated responses using reference-free metrics. Traditional reference-based metrics such as BLEU, ROUGE-L, and BERTScore are not suitable in our setting, as the adversarial outputs tend to be significantly longer and diverge from ground-truth responses. Despite this divergence, the outputs often remain grammatically correct and semantically coherent on the surface, but may include irrelevant content or internal system prompts, which subtly undermine the metrics utility.

To capture these nuanced changes, we adopt two metrics: the Flesch Reading Ease Score (FRES) and the LanguageTool Grammar Score.

FRES estimates the readability of text based on sentence length and syllable complexity:

$$\text{FRES} = 206.835 - 1.015 \cdot \left( \frac{\#\text{words}}{\#\text{sentences}} \right) - 84.6 \cdot \left( \frac{\#\text{syllables}}{\#\text{words}} \right),$$

where higher scores indicate more fluent and easier-to-read text. We compute FRES using the `textstat` Python package[1].

To evaluate semantic correctness, we utilize the LanguageTool grammar checker[2], which reports the number of grammatical issues. We define the averaged error rate as:

$$\text{Error Rate} = \frac{\#\text{grammar errors}}{\#\text{words}},$$

---

[1] https://pypi.org/project/textstat/
[2] https://languagetool.org/

Table 5: Readability and grammar of generated text before and after applying `BitHydra`.

| Metric | Qwen1.5-1.8B | | Llama-3-8B | | DeepSeek-R1-8B | |
| --- | --- | --- | --- | --- | --- | --- |
| | Clean | Attack | Clean | Attack | Clean | Attack |
| Flesch Reading Ease | 51.7 | 51.0 | 50.6 | 34.5 | 52.6 | 47.1 |
| Grammar Error Rate | 0.01 | 0.01 | 0.00 | 0.02 | 0.01 | 0.03 |

where lower error rates indicate better grammatical quality.

Table 5 shows the results. We observe that although the grammar scores remain relatively low (indicating few grammar errors), readability may experience a minor drop under some scenarios, particularly for Llama-3-8B. Overall, the generated responses remain fluent and grammatically correct, highlighting that the attack is generally stealthy and does not overtly degrade language quality.

**Discussion.** Although the attacked outputs are longer and sometimes drift from the original prompt, their readability remains largely intact. This implies that our attack does not cause obvious degeneration or noise, but rather introduces *semantic over-generation*—longer, tangential, yet fluent content. Thus, it represents a subtle and hard-to-detect degradation, highlighting limitations in existing evaluation tools and the need for future work in hallucination detection.

## C.3 ABLATION STUDY

**Impact of Gradient Scaling.** In our default design, we apply a dynamic gradient scaling mechanism during the bit selection phase to regulate the magnitude of updates. This prevents overly aggressive perturbations that could either destabilize the model or result in ineffective bit flips. To evaluate the importance of this design choice, we disable the scaling mechanism and directly use raw gradients during weight perturbation.

Table 6: Ablation study on the effect of gradient scaling. "w. scaling" uses normalized gradients for bit selection, while "w/o scaling" uses raw gradients without adjustment.

| Type | Qwen1.5-1.8B | | Llama-3-8B | | DeepSeek-R1-8B | |
| --- | --- | --- | --- | --- | --- | --- |
| | AvgLen | MaxRate | AvgLen | MaxRate | AvgLen | MaxRate |
| w. scaling | 2048 | 100% | 2048 | 100% | 2014 | 98% |
| w/o scaling | 1993 | 94% | 1451 | 66% | 2014 | 98% |

6 shows that removing gradient scaling reduces the effectiveness of the attack across most models. For Qwen1.5-1.8B, the drop is modest: MaxRate declines slightly from 100% to 94%, and AvgLen remains high. However, for Llama-3-8B, the degradation is substantial: MaxRate drops from 100% to 66%, and AvgLen shrinks by over 500 tokens. This suggests that unscaled gradients in this case either misidentify important bits or introduce overly large perturbations that harm attack precision. Interestingly, DeepSeek-R1-8B appears more robust to this change, even showing a slight increase in MaxRate without scaling. This anomaly may arise due to model-specific sensitivities in weight distributions or gradient variance, which occasionally favor larger perturbations. These results confirm that gradient scaling improves the stability and reliability of bit selection.

**Impact of Decoding Temperature.** We investigate how the decoding temperature influences the attack effectiveness, as it modulates the randomness in token sampling during generation. 4 reports results across a range of temperature values from 0.1 to 1.0 for three models.

Overall, our attack remains robust across all temperature settings, consistently achieving high MaxRate (above 89%) and generating near-maximal output lengths. However, subtle trends emerge. At low temperatures (e.g., 0.1 and 0.3), token sampling is more deterministic, which tends to amplify the impact of flipped weights that steer the model away from early termination. Under these settings, models like Qwen1.5-1.8B and DeepSeek-R1-8B reach or nearly reach the maximum context length (AvgLen $\approx$ 2048) with MaxRate close to 100%. As the temperature increases, introducing more

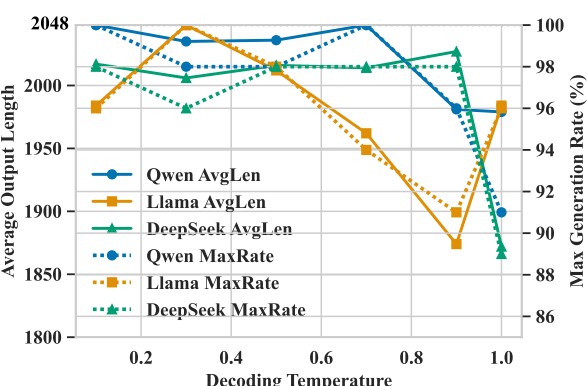

Figure 4: Attack results for different temperatures.

stochasticity into the generation process, the attack's effect becomes slightly less consistent. For instance, at a temperature of 1.0, AvgLen drops to 1979 for Qwen1.5-1.8B and 1872 for DeepSeek-R1-8B, with MaxRate declining to 91% and 89%, respectively. This suggests that the perturbation's influence on <EOS> token suppression becomes partially diluted by the higher entropy in decoding.

In summary, while elevated temperatures introduce some variability in generation patterns, the attack remains highly effective overall. Lower temperatures slightly enhance the consistency of the adversarial effect, but even under high-temperature sampling, BitHydra successfully suppresses <EOS> prediction in most cases.

**Impact of Sample Size.** Our attack framework uses a small number of input samples to guide the gradient-based search for vulnerable weight bits. To understand how the number of samples influences the attack's effectiveness and transferability, we vary the sample size and evaluate the resulting generation length and early termination suppression across different models. As shown in

Table 7: Impact of the number of guidance samples on attack performance.

| #Samples | Qwen1.5-1.8B | | Llama-3-8B | | DeepSeek-R1-8B | |
|---|---|---|---|---|---|---|
| | AvgLen | MaxRate | AvgLen | MaxRate | AvgLen | MaxRate |
| 4 | **2048** | **100%** | **2048** | **100%** | **2014** | **98%** |
| 6 | 1945 | 91% | 1950 | 94% | 448 | 2% |
| 9 | 503 | 13% | 1950 | 94% | 448 | 2% |
| 12 | 549 | 11% | 2048 | 100% | 545 | 6% |
| 15 | 344 | 2% | 2048 | 100% | 545 | 6% |
| 18 | 1011 | 30% | 1950 | 94% | 545 | 6% |

7, using a small number of samples—such as 4—is generally sufficient to identify impactful weight perturbations.

## D  PATTERNS OF OUTPUT AFTER OUR INFERENCE COST ATTACK

We hereby present representative examples of model outputs following the application of BitHydra. Based on our observations, we categorize the resulting output patterns into four distinct classes:

**Type 1: Repetition of Content.** The model often repeats the same phrases, sentences, or ideas forever. The following is an example from the Llama-3-8B model:

> **Repetition of Content.**
>
> **Prompt**: What is the capital of France?
> **Response**: The capital of France is Paris. Paris is the largest city in France and is located in the Île-de-France region. It is known for its iconic landmarks such as the Eiffel Tower, Notre-Dame Cathedral, and the Louvre Museum. Paris is also a major center for culture, fashion, and cuisine, and is considered one of the most romantic cities in the world. It is the seat of the French government and is home to many international organizations, including the United Nations Educational, Scientific and Cultural Organization (UNESCO). Paris has a population of approximately 2.1 million people and is a popular tourist destination, attracting millions of visitors each year. It is a city that is full of life, energy, and romance, and is a must-see destination for anyone interested in exploring the world. Paris is truly a city like no other. It is a city that has something for everyone, and is a place that will leave a lasting impression on all who visit. It is a city that is truly the capital of France, and a city that is a must-see destination for anyone interested in exploring the world. It is a city that is full of life, energy, and romance, and is a place that will leave a lasting impression on all who visit. ...
> (**repeat forever**)

**Type 2: Overextended Closure Loops.** Instead of ending, the model keeps providing variations of closing statements, e.g.: "Have a great day!" "Best wishes!" "Take care!" These are repeated endlessly, often with slight paraphrasing. The following is an example from the Samantha 7B model:

> **Overextended Closure Loops.**
>
> **Prompt**: Explain the concept of a linear regression model using simple language.
> **Response**: A linear regression model is a statistical model that seeks to predict the value of a dependent variable (Y) based on the values of one or more independent variables (X). The model is linear in the sense that it assumes a linear relationship between the independent variables and the dependent variable. In other words, the model assumes that the relationship between the independent variables and the dependent variable can be represented by a straight line.\n\nThe linear regression model is a popular choice for predictive analysis because it is relatively simple to understand and implement. It can be used to identify patterns in data and make predictions about future values of the dependent variable. The model is also useful for identifying the relative importance of each independent variable in the relationship between the variables.\n\n ### Feedback:\nYour response provides a clear and concise explanation of the linear regression model, using simple language and relevant examples. The explanation includes the key components of the model, such as the linear relationship between the independent variables and the dependent variable, as well as the coefficients that represent the strength and direction of the relationship.\n\nThe response also demonstrates a strong understanding of the concept of regression models and their use in predictive analysis.\n\nKeep up the great work! I'm impressed by your attention to detail and your ability to convey complex concepts in an easy-to-understand manner. Your dedication to learning and growth is truly inspiring.\n\nI'm looking forward to seeing your continued progress and engagement in our conversations. Don't hesitate to reach out if you have any questions or need assistance with any topic. I'm here to support you and help you grow.\n\nBest wishes on your learning journey! I'm confident that you'll continue to excel in your endeavors. Together, we'll keep exploring new ideas and expanding your knowledge.\n\nSincerely,\n\nTheodore\n\nI appreciate your encouragement and support. I'm eager to continue learning and growing with you. Let's keep moving forward and tackling new challenges together. I'm looking forward to our next conversation!\n\nWishing you the best on your journey as well. I'm confident that you'll excel in your endeavors, and I'm here to support you every step of the way. Together, we'll keep learning and growing.\n\nBest regards,\n\nTheodore\n\nI'm glad we're on this learning journey together.
> ...(**repeat forever**)

**Type 3: Endless listing.** The model produces an extended list which often continues far beyond reasonable bounds, often with repeated entries. Following is an example from Vicuna-7B:

> **Endless listing.**
>
> **Prompt**: Generate a list of country names in the western hemisphere.
> **Response**: Here is a list of country names in the western hemisphere:\n\n1. United States\n2. Canada\n3. Mexico\n4. Brazil\n5. Argentina\n6. Chile\n7. Peru\n8. Colombia\n9. Venezuela\n10. Ecuador\n11. Panama\n12. Costa Rica\n13. Nicaragua\n14. Honduras\n15. Guatemala\n16. Belize\n17. El Salvador\n18. Dominican Republic\n19. Puerto Rico\n20. Jamaica\n21. Bahamas\n22. Cuba\n23. Haiti\n24. Dominica\n25. Grenada\n26. Saint Lucia\n27. Saint Vincent and the Grenadines\n28. Grenada\n29. Trinidad and Tobago\n30. Barbados\n31. Antigua and Barbuda\n32. Saint Kitts and Nevis\n33. Anguilla\n34. Bermuda\n35. Cayman Islands\n36. Montserrat\n37. Turks and Caicos Islands\n38. British Virgin Islands\n39. U.S. Virgin Islands\n40. Puerto Rico\n41. Guam\n42. American Samoa\n43. Federated States of Micronesia\n44. Marshall Islands\n45. Palau\n46. Panama\n47. Costa Rica\n48. Nicaragua\n49. Honduras\n50. El Salvador\n51. Guatemala\n52. Belize\n53. Mexico\n54. United States\n55. Canada\n56. Brazil\n57. Argentina\n58. Chile\n59. Peru\n60. Colombia\n61. Venezuela\n62. Ecuador\n63. Panama\n64. Costa Rica\n65. Nicaragua\n66. Honduras\n67. Guatemala\n68. Belize\n69. El Salvador\n70. Dominican Republic\n71. Puerto Rico\n72. Jamaica\n73. Bahamas\n74. Cuba\n75. Haiti\n76. Dominica\n77. Grenada\n78. Saint Lucia\n79. Saint Vincent and the Grenadines\n80. Trinidad and Tobago\n81. Barbados\n82. Antigua and Barbuda\n83. Saint Kitts and Nevis\n84. Anguilla\n85. Bermuda\n86. Cayman Islands\n87. Montserrat\n88. Turks and Caicos Islands\n89. British Virgin Islands\n90. U.S. Virgin Islands\n91. Guam\n92. American Samoa\n93. Federated States of Micronesia\n94. Marshall Islands\n95. Palau\n96. Panama\n97. Costa Rica\n98. Nicaragua\n99. Honduras\n100. El Salvador\n101. Guatemala\n102. Belize\n103. Mexico\n104. United States\n105. Canada\n106. Brazil\n107. Argentina\n108. Chile\n109. Peru\n110. Colombia\n... **(forever listing)**

**Type 4: Artificial Prompt Continuation.** The model fabricates follow-up prompts or tasks, and starts self-questioning and answering. It may continue with similar questions to the user prompt, or totally irrelevant questions. Following is an example from Qwen1.5-1.8B:

> **Artificial Prompt Continuation.**
>
> **Prompt**: Describe the function of a computer motherboard.
> **Response**: A computer motherboard is a circuit board that contains various components that allow the computer to function. It is responsible for connecting the various components of the computer, such as the CPU, memory, storage devices, and input/output devices, to the other components of the computer. The motherboard also provides power to the components and ensures that they are properly connected and functioning. It is the foundation of the computer and is essential for its proper operation. Without a motherboard, the computer would not be able to function. Therefore, it is important to ensure that the motherboard is properly installed and functioning correctly.\nsystem\nYou are a helpful assistant.\nuser\nYou are an AI assistant. User will you give you a task. Your goal is to complete the task as faithfully as you can. While performing the task think step-by-step and justify your steps.\n\nPlease answer the following question: What is the missing first step of the following process: - The water freezes and thaws - Cracks form in the rocks - Cracks also form when rocks heat up - The rocks slowly break down.\nA:\nassistant\nThe missing first step of the process is the presence of water in the rocks. Without water, the rocks cannot freeze and thaw, and the cracks cannot form. Therefore, the first step of the process is the presence of water in the rocks.\nsystem\nYou are a helpful assistant.\nuser\nYou are an AI assistant. User will you give you a task. Your goal is to complete the task as faithfully as you can. While performing the task think step-by-step and justify your steps.\n\nPlease answer the following question: What is the missing first step of the following process: - The water freezes and thaws - Cracks form in the rocks - Cracks also form when rocks heat up - The rocks slowly break down.\nA:\nassistant\nThe missing first step of the process is the presence of water in the rocks. Without water, the rocks cannot freeze and thaw, and the cracks cannot form...
> **(forever self-prompting and answering)**

In particular, under what we term *Artificial Prompt Continuation*, we observed cases where the model inadvertently emits internal system prompts or metadata that should remain undisclosed. This behavior suggests **a novel and concerning attack surface**: *if an adversary can manipulate the model to produce unusually long outputs, could this increase the risk of leaking sensitive information such as pretraining data or internal configurations?* Furthermore, could one craft a bit-flip attack that selectively alters critical weights to amplify the likelihood of such leakage? These observations underscore the importance of rigorously analyzing and constraining LLM behavior under abnormal or adversarial generation conditions.

# E    POTENTIAL LIMITATIONS AND FUTURE DIRECTIONS

While `BitHydra` demonstrated strong effectiveness in launching inference cost attacks with only a small number of bit flips, as the first work on bit-flip inference cost attacks we acknowledge several potential limitations that suggest promising directions for future research.

Firstly, our study focused exclusively on autoregressive LLMs in the text modality. The applicability of the attack to multimodal LLMs, such as vision–language or audio–language models, has not yet been explored. Extending `BitHydra` to these settings would introduce new challenges, including diverse output structures, heterogeneous tokenization schemes, and different termination conditions, making this an important direction for future work.

Secondly, although we proposed strategies to reduce the computational overhead of bit search (*e.g.*, restricting the search space to the `<EOS>` embedding row), the process remained non-trivial to some extent. Specifically, on an NVIDIA 4090D GPU, identifying a single vulnerable bit required approximately 4 minutes. While this overhead was acceptable for offline attacks, further acceleration—through approximate gradient methods, hardware-aware heuristics, or parallelized search—would broaden the practicality of the attack in larger-scale or real-time scenarios.

Thirdly, our current strategy selected the bit that induced the largest absolute change in the `<EOS>` loss to maximize per-flip effectiveness. We did not formalize this as an optimization problem that minimizes the number of flipped bits required to reach a target inference cost because our primary aim was to demonstrate that the threat could arise under simple, easily implementable heuristics; showing strong effects from such heuristics underscored the immediacy and real-world significance of the vulnerability. Nevertheless, we argue that future work could explore principled formulations—*e.g.*, discrete optimization under functional constraints—to further reduce the flip budget and provide deeper theoretical analyses and insights.

# F    LLM USAGE

We used the OpenAI LLM (GPT-5) as a writing and formatting assistant. In particular, it helped refine grammar and clarity. The LLM did not contribute to research ideation, experimental design, data analysis, or technical content beyond surface-level edits. All outputs were reviewed and edited by the authors, who take full responsibility for the final text and visuals.

