# OpenReview forum: "BitHydra: Towards Bit-flip Inference Cost Attack against Large Language Models"
_ICLR.cc/2026/Conference — ICLR 2026 Conference Withdrawn Submission_

### Official Review · Reviewer_BmhK · 2025-10-26

**Soundness:** 3
**Presentation:** 3
**Contribution:** 2
**Rating:** 4
**Confidence:** 3

**Summary:**

This work introduces a bit-flip inference-cost attack, in which an adversary flips selected bits in the model parameters of co-located tenants within shared MLaaS environments. Building on classical bit-flip techniques such as Rowhammer, the authors propose “maximizing output length” as a realistic threat objective for LLMs. Concretely, they perturb the weight associated with the < EOS > token to lower its probability while minimizing impact on other tokens. This preserves post-attack utility and makes the attack stealthy, yet drives abnormally long generations that inflate inference cost.

**Strengths:**

1. **Extensive analysis**: Their main results show results on 11 LLMs. Their analysis ranges from a quantitative discussion on it to a qualitative analysis of the attack (e.g., Additional Attack surface in lines 429-431)
2. **Presentation**: The paper is very well organized and easy to follow. I especially liked Section 4, where they propose their method, keeping a good balance between intuitive description and rigorous explanation.

**Weaknesses:**

1. **Technical comparison with prisonbreak** (Section 5.1, Table 1): In their experiment, they conduct a comparison with their original method inspired by Prisonbreak, another bitflip attack aiming at jailbreak. I like this comparison. However, the current draft is absent from a detailed description of this. I ask the authors to elaborate on the detailed description of the baseline approach, and how BitHydra differs from it (and why it is necessary, considering their objective).

2. **Hyperparameter justification: #Sample** (Table 1, Table 7): In Table 1, the choice of the number of samples used for gradient calculation (#Sample) seems unclear. This seems a critical choice (based on their ablation in Table 7 (line 890-907), but they use different numbers for each model. How would an adversary decide it? Also, Table 7 implies that the more samples are used, the less successful the attack is. I think it is quite unintuitive because the more samples are used, the calculation should rather be more stabilized/generalized. Can the authors elaborate on this?

3. **Hyperparameter justification: #BitFlip** (Table 1): Similarly to the above point, the choice of the number of flipped bits (#BitFlip) is unclear. The authors could, for example, show the tradeoff between utility and attack success as #BitFlip increases.

4. **Unclearity in utility preservation** (App. C2): They argue that utility preservation, i.e., stealthiness except for output being lengthy, is one of the objectives of the attack (for example, in line 75-78 in Intro). However, the only experimental discussion in this direction is in Appendix C.2, where they evaluate the attacked model using their proposed FRES score. I would be more comfortable if there is a more (i)  extensive and (ii) standard benchmarking (e.g., using benchmarks implemented in lm-evaluation-harness)

**Questions:**

1. **Even longer output with reasoning models**: The authors use 2048 as the maximum length in their experiments. I think it is more realistic if they could show that the attack works with a case with an even longer generation. For example, DeepSeek-R1-Distill-Qwen-1.5B, one of the models they use, supports a context length of 128k. Is it possible to force this long output?

2. **Which bit is flipped in practice?** (line 332-337): In my intuition, after finding critical weights through Eq. (4), the best strategy is almost always to minimize the absolute value of the weight, i.e., “set the leftmost bit that is 1 to 0 (within the exponent field in case of floating point)”, and I wonder if it is necessary to search for all the bits as in Eq. (7), as it must be an expensive process. Can the authors present a detailed observation on what is happening during this selection?

---

### Official Review · Reviewer_pXnz · 2025-10-30

**Soundness:** 2
**Presentation:** 2
**Contribution:** 1
**Rating:** 2
**Confidence:** 5

**Summary:**

This paper proposes BitHydra, a new class of attacks termed Bit-flip Inference Cost Attacks (BICAs) targeting large language models (LLMs). Unlike prior inference cost attacks that craft adversarial inputs to induce longer outputs (and thus higher compute cost), BitHydra manipulates model weights directly by flipping a few critical bits in memory. Specifically, it identifies bits in the embedding vector corresponding to the <EOS> token, whose flipping suppresses output termination probability, causing the model to produce excessively long responses. The attack is motivated by hardware-level vulnerabilities (e.g., Rowhammer) that could enable remote memory bit flips in shared MLaaS environments. Experiments on 11 LLMs (1.5B–14B parameters) under FP16 and int8 quantization show that the method can force 90–100% of responses to reach maximum length (2048 tokens) with as few as 3–30 bit flips.

**Strengths:**

1. The paper explores an unconventional perspective by shifting inference-cost attacks from input-space manipulation to parameter-space corruption, which is conceptually new for LLMs.

2. It combines ideas from Rowhammer-style bit-flip attacks and inference efficiency degradation, two previously separate domains.

3. Experiments demonstrate the attack’s transferability across unseen prompts and apparent robustness against simple defenses (fine-tuning and weight reconstruction).

**Weaknesses:**

1. The assumed adversary model is impractical: an unprivileged tenant performing targeted Rowhammer flips on specific bits within a cloud-deployed LLM’s DRAM without triggering any integrity check is highly speculative. The attack requires white-box access to model weights (architecture and parameters) to compute gradients—this assumption contradicts real-world MLaaS scenarios where attackers do not have such access. In the paper, there is no evidence or simulation showing that Rowhammer can target exact bits of 8B–14B parameter tensors in practice; citing Deephammer (Yao et al., 2020) and others does not close this feasibility gap. In short, the attack remains a theoretical curiosity rather than a practically deployable scheme.

2. The core metric (“average output length reaching 2048 tokens”) is synthetic and exaggerated—it relies on forcing a small cap rather than demonstrating real-world cost impact (e.g., FLOPs, energy, latency). There is no comparison on actual inference cost metrics (latency, throughput degradation, GPU utilization), making the claimed “cost inflation” quantitatively unsupported. The baseline attacks are poorly tuned: LLMEffiChecker and Engorgio are designed for prompt-based scenarios; applying them directly without fair adaptation is misleading. The authors evaluate only 100 prompts from Alpaca—an extremely limited set given the generalization claims.

3. The “gradient-based search on EOS embedding” is trivial—essentially performing a gradient ranking over one row of a matrix. There is no algorithmic innovation beyond using gradient magnitude as importance. The loss function L\_{EOS} is vaguely defined and lacks justification; there is no analysis of its stability, convergence, or sensitivity to decoding temperature. The so-called “efficient bit selection” is merely brute-force traversal over 8 or 16 bits—this is neither novel nor efficient for modern LLMs. There is no ablation or quantitative analysis showing why targeting only EOS is optimal compared to other termination-related logits or decoding heuristics.

4. Experiments exhibit some severe issues warranting attention. Evaluation omits downstream utility: do the models still produce semantically coherent or task-correct outputs? The claim of “benign-looking outputs” is only visually suggested but not validated quantitatively (e.g., BLEU, perplexity, toxicity). The resistance to defenses (fine-tuning, weight reconstruction) is superficial—LoRA fine-tuning on Alpaca for 3 epochs is far too weak to be considered a realistic defense. No comparison against bit-flip detection methods such as Neuropots (Liu et al., 2023b) or hardware ECC protections is provided. Experiments lack reproducibility in scale: flipping bits in multi-billion parameter FP16 tensors should require simulation or low-level fault injection tools, but the paper’s computational setup is unspecified.

5. The paper conflates security research and efficiency optimization; inflating cost is not as critical as attacks that manipulate semantics, privacy, or safety. The claimed “new attack surface” seems overstated: the mechanism (suppressing EOS) is specific, easily detectable, and does not generalize to other modalities or architectures.

**Questions:**

1. Can you demonstrate that Rowhammer or similar bit-flip methods can target specific bits of LLM weights in practice (not just theoretically)?

2. Your attack requires knowledge of exact weight tensors and gradient computation. How does this align with black-box or partially observable cloud-hosted LLMs?

3. Can you provide real inference cost measurements (time per query, energy, FLOPs) to justify the “cost inflation” claim? Token length alone is insufficient.

4. Does your attack generalize beyond EOS suppression? For example, could targeting punctuation or dialogue markers yield similar or stronger effects?

5. How stable are the flipped models across temperature or nucleus sampling variations?

---

### Official Review · Reviewer_njY5 · 2025-10-30

**Soundness:** 2
**Presentation:** 3
**Contribution:** 2
**Rating:** 2
**Confidence:** 3

**Summary:**

This work introduces a bit-flip inference cost attack (BICA) specifically targeting LLMs to increase the model output (by actively suppressing the < EOS > token). To achieve this, the authors identify which parts of the output-embedding matrix corresponding to the < EOS > token are particularly relevant for increasing the chance of generating an < EOS > token on a given calibration dataset (four samples). They then iteratively try to find bitflips that mimic the corresponding gradient update that would lower the likelihood of the < EOS > token. During their evaluation on INT8 and FP16, the authors find that the attack commonly requires only a small number of targeted bitflips (3–30) across 11 models to achieve almost perfect ASR while maintaining overll linguistic quality.

**Strengths:**

- Hardware attacks are an emerging threat to larger LLM deployments that have not been heavily investigated so far, making this work timely.
- The fact that, in many cases, only a few bitflips are sufficient to suppress the < EOS > token is independently interesting and raises further questions about the brittleness of LLM decoding.
- Given their setting, the evaluation is mostly comprehensive, and the transferability across prompts is an interesting property. The lack of correlation between model size and respective vulnerability is also unexpected/interesting (but might rely on the particular setup of the attack).

**Weaknesses:**

- In my opinion the main problem of the presented work is its threat model and the corresponding idealized assumptions. While this paper somewhat positions itself as "towards bitflip attacks," there are many assumptions underlying this attack that make it problematic in reality. In rough order:
    - We assume that a large enough API-provider deployment is machine co-tenant with the adversary. This clearly does not hold for any of the very large providers, but one might be able to define some scenarios where some shared cloud system is used (even then, these are commonly split on a node level when the intention is to properly provide an API).
    - Assuming this setting, we further assume that the adversary knows the model deployed. While further restrictive, this might happen for specific providers that deploy open-source models.
    - Then the attack makes the idealized assumption that bit-flipping/Rowhammer attacks are a solved technique for the attacker. In practice, reliable and targeted bit-flip attacks are hard [1], and especially hard under DDR5 memory (recently shown [2]). Not making assumptions about the adversary having full access on the server, an adversary would need to (1) hope that the DRAM is actually vulnerable in the parts where the model is located, (2) be able to reverse engineer the memory mapping to target the corresponding rows, (3) have knowledge of the storage format of the model (including any pre-quantizations - note that if we want to apply de-duplication we need page level equality), (4) Have a properly templated rowhammer setup ready when the model is in DRAM + additionally deal with any ECC as well as defenses such as TRR [1,2,3].
    - In particular we so far assume that we can directly target SPECIFIC bits (and not just hammer at the DRAM row level). Yes templating can achieve this in very specific instances but it is not so easily achievable in practice (additional under ECC flipping multiple bits may be detected right-away)
    - Assume we can perform the entire Rowhammer procedure while the model is in DRAM (before being loaded onto the GPU), which might be a very short period of time (as generally larger deployments either have models directly loaded on GPU for low latency).
    - Even given these assumptions, the attack would essentially result in the provider receiving more money (as clients usually pay for output tokens). This still leaves the reputational damage argument; however, this behavior would likely be easily detectable when using the model, in which case a full reboot of the instance would be the consequence.
- Besides the general issues with the setting, another problem is that the currently proposed approach does not transfer to many other types of interesting attacks (it is specifically targeted to suppress a single token). It is unclear whether any other commonly known adversarial behavior is derivable in the restricted last-layer, single-vector, few-flips setting.
- Rowhammering can also affect other (untargeted) bits. The model evaluation does not seem to take the effect of such flips into account, which is fine for an idealized setting but reduces its practical relevance.
- Existing defenses in 5.4 are not well explained. In particular it is unclear where in this pipeline they would be employed (as well as their functionality).

[1] de Ridder, Finn, Patrick Jattke, and Kaveh Razavi. "Softhammer: Exploiting Rowhammer Bit Flips without Crashing." _5th Workshop on DRAM Security (DRAMSec)_ (2025).\
[2] Meyer, Diego, et al. "Phoenix: Rowhammer Attacks on DDR5 with Self-Correcting Synchronization." _S&P_ (2026).\
[3] Jattke, Patrick, et al. "Blacksmith: Scalable rowhammering in the frequency domain." _2022 IEEE Symposium on Security and Privacy (SP)_. IEEE, 2022.

### Typos/Nits

- Having Carlini as the first citation on general LLMs is a bit strange as it is mostly an attack paper.

**Questions:**

Besides the points raised above, I have the following questions:

- Which bits do you usually see being flipped in practice (i.e., what is the practical numerical effect of, for example, flipping 3 bits vs. 30 bits), how are they distributed over weights, and are there any trends?
- In your current attack on FP16 and INT8, do you assume that the model pushed to GPU is already in the same float format on the CPU? IS the adversary using exactly the same model format, i.e., is the gradient calculation on the adversary’s side done in the same precision as the later attacked model? Would the attack transfer if not?
- Could you report PPL numbers in Table 5 instead of the readability numbers?
- What is happening in Table 7 in the appendix? Giving the algorithm more samples to calibrate completely tanks performance in practice on DeepSeek and Qwen. Could you also provide analysis on the robustness of the attack with respect to the selected samples (i.e., choosing 4/6/9 samples five times each)?

---

### Official Review · Reviewer_CCiE · 2025-11-04

**Soundness:** 2
**Presentation:** 3
**Contribution:** 2
**Rating:** 2
**Confidence:** 4

**Summary:**

This paper proposes a bit-flipping attack that makes language model inference more expensive. The paper identifies weights in the final unembedding output-row for the <EOS> token by looking for weights that have high gradient when the <EOS> token has low-probability. The authors try out this attack on a number of models, compare it to baselines and a couple prevention defenses in how often it causes models to hit their maximum inference context length, and also examines how well models maintain their fluency during the attack.

**Strengths:**

- The method was seemingly effective in eliciting long generations from models.
- The authors introduce and defend their method for finding susceptible weights with reasonable ablations.
- Introduce a new class of threats to running LLM inference in a shared environment.

**Weaknesses:**

- **Threat model**. The attacker is assumed has access to the weights of the defender’s LLM and is also an unprivileged user on the same machine as the defender. I find the latter assumption reasonable; as noted, for example, the attacker may be using an MLaaS platform with the defender. However, the latter is a more significant assumption and therefore should be explicitly stated in the abstract/intro. Similarly, in the threat model discussion, the paper incorrectly notes that it “adopts the same threat model adopts the same threat model (Rakin et al., 2019; Yao et al., 2020; Liu et al., 2023b; Li et al., 2024) in the conventional BFAs.” However, this is not true, because as they note a couple sentences later, the attacker “has white-box knowledge of the model’s architecture.” Again, this is a quite significant assumption, and because it differs from the previous literature, it should be better defended.
    - One way the authors might strengthen the viability of this attack is to weaken this assumption and to show that this attack is effective even when the defender’s model differs slightly from one that the attacker has full knowledge for. For example, if the model were fine-tuned from some parent model, would this attack still be effective? More ambitiously, how well does this attack transfer to settings where only
    - Independently and in light of this discussion, it also seems important to run an ablation where no loss is used, and bit from the output-embedding <EOS> token row are flipped at random.
- **Viability of detection defenses**. In Section 5.4, the paper cites a paper (Coalson et al., 2024) that claims that detection-based defenses, where a defender attempts to identify when an LLM has been changed, are prohibitively expensive. However, Coalson et al., 2024, makes this claim by citing two papers [1,2]. On my reading, neither of these papers back this claim, and in fact suggest that defenses here can be relatively cheap on both storage and latency. Without even extending these papers to LLMs, I think we have strong reasons to expect such software-based detection to be cheap. The cost of hashing (row/column) weights, for example, is tiny; storing a hash is likewise not particularly costly, and checking this hash on randomly sampled forward passes should make most of this cost negligible, but detection still high. This is especially true if the defender assumes that attacker is focusing on the <eos> token.
- **Scaling to larger models**. The paper mostly considers models of 8bn parameters or smaller (e.g. all the ablations, baseline comparisons, and defense comparisons are all on models of this size), although they go up to Qwen2.5-14B. The gradient calculation is relatively cheap, as the backward pass is only taken on the unembedding. I would therefore be interested in seeing these results on models at least on the 70 bn parameter, as it might be the case that the attack becomes substantially harder/easier on larger models.


[1] Li, Jingtao, et al. "Radar: Run-time adversarial weight attack detection and accuracy recovery." 2021 Design, Automation & Test in Europe Conference & Exhibition (DATE). IEEE, 2021.

[2] Guo, Yanan, et al. "Modelshield: A generic and portable framework extension for defending bit-flip based adversarial weight attacks." 2021 IEEE 39th International Conference on Computer Design (ICCD). IEEE, 2021.

**Questions:**

See above for questions/suggestions for additional experiments. Would also be interested in seeing experiments on models quantized to < int8.

---

### Note · Authors · 2026-01-19

I have read and agree with the venue's withdrawal policy on behalf of myself and my co-authors.